# Seroepidemiology of SARS-CoV-2 in a cohort of pregnant women and their infants in Uganda and Malawi

Lauren Hookham[1]*, Liberty Cantrell[2], Stephen Cose[3,4], Bridget Freyne[5,6,7], Luis Gadama[8], Esther Imede[4], Kondwani Kawaza[8], Samantha Lissauer[5,6], Phillipa Musoke[9], Victoria Nankabirwa[10], Musa Sekikubo[11], Halvor Sommerfelt[12,13], Merryn Voysey[2], Kirsty Le Doare[1,14‡], on behalf of The periCOVID Consortium[¶]

1 Institute for Infection and Immunity, St George's, University of London, London, United Kingdom, 2 Oxford Vaccine Group, Department of Paediatrics, University of Oxford, Oxford, United Kingdom, 3 London School of Hygiene & Tropical Medicine, London, United Kingdom, 4 MRC/UVRI and LSHTM Uganda Research Unit, Entebbe, Uganda, 5 Malawi Liverpool Wellcome Trust, Blantyre, Malawi, 6 Institute of Infection, Veterinary and Ecological Science, University of Liverpool, Liverpool, United Kingdom, 7 School of Medicine, University College Dublin, Dublin, Ireland, 8 Kamuzu University of Health Sciences, Blantyre, Malawi, 9 Department of Paediatrics and Child Health, Makerere University, Kampala, Uganda, 10 Department of Epidemiology & Biostatistics, School of Public Health, Makerere University, Kampala, Uganda, 11 Department of Obstetrics & Gynaecology Makerere University, Kampala, Uganda, 12 Centre for Intervention Science in Maternal and Child Health, Department of Global Public Health and Primary Care, University of Bergen, Bergen, Norway, 13 Norwegian Institute of Public Health, Bergen, Norway, 14 Makerere University, John's Hopkins University, Kampala, Uganda

‡ KLD is senior authorship on this work.
¶ Membership of the periCOVID Consortium is provided in the Acknowledgments.
* lhookham@sgul.ac.uk

**Data Availability Statement:** The data relevant to this study are available at https://doi.org/10.24376/rd.sgul.24864519.v1. The data dictionary is

## Abstract

### Background

Data on SARS-CoV-2 infection in pregnancy and infancy has accumulated throughout the course of the pandemic, though evidence regarding asymptomatic SARS-CoV-2 infection and adverse birth outcomes are scarce. Limited information is available from countries in sub-Saharan Africa (SSA). The pregnant woman and infant COVID in Africa study (PeriCOVID Africa) is a South-South-North partnership involving hospitals and health centres in five countries: Malawi, Uganda, Mozambique, The Gambia, and Kenya. The study leveraged data from three ongoing prospective cohort studies: Preparing for Group B Streptococcal Vaccines (GBS PREPARE), SARS-CoV-2 infection and COVID-19 in women and their infants in Kampala and Mukono (COMAC) and Pregnancy Care Integrating Translational Science Everywhere (PRECISE). In this paper we describe the seroepidemiology of SARS-CoV-2 infection in pregnant women enrolled in sites in Uganda and Malawi, and the impact of SARS-CoV-2 infection on pregnancy and infant outcomes.

### Outcome

Seroprevalence of SARS-CoV-2 antibodies in maternal blood, reported as the proportion of seropositive women by study site and wave of COVID-19 within each country.

available at https://doi.org/10.24376/rd.sgul.
22116965.v1.

**Funding:** This study was funded by The European
and Developing Countries Clinical Trials
Partnership (EDCTP) (grant number RIA2020EF-
2926 periCOVID Africa) and the Research Council
of Norway (grants 312768 and 223269). The
funders had no role in study design, data collection
and analysis, decision to publish, or preparation of
the manuscript. VN is supported by an EDCTP2
Senior Fellowship (TMA2018SF-2479).

**Competing interests:** The authors have declared
that no competing interests exist.

## Methods

The PeriCOVID study was a prospective mother-infant cohort study that recruited pregnant women at any gestation antenatally or on the day of delivery. Maternal and cord blood samples were tested for SARS-CoV-2 antibodies using Wantai and Euroimmune ELISA. In peri-COVID Uganda and Malawi nose and throat swabs for SARS-Cov-2 RT-PCR were obtained.

## Results

In total, 1379 women were enrolled, giving birth to 1387 infants. Overall, 63% of pregnant women had a SARS-CoV-2 positive serology. Over subsequent waves (delta and omicron), in the absence of vaccination, seropositivity rose from 20% to over 80%. The placental transfer GMR was 1.7, indicating active placental transfer of anti-spike IgG. There was no association between SARS-CoV-2 antibody positivity and adverse pregnancy or infancy outcomes.

## Introduction

The initial predictions of the impact of SARS-CoV-2 in sub-Saharan Africa suggested high case numbers and fatalities [1], yet there is evidence to suggest that the pandemic evolved differently in Africa than in other regions [2, 3]. Several factors have been proposed to explain the relatively low frequency of severe SARS-CoV-2 illness, including a younger population, a lack of long-term care facilities and reduced population density [4–7]. However, limited testing capacity and weak reporting structures in many sub-Saharan African countries may also result in under-reporting, leading to an underestimation of the true risk of serious SARS-CoV-2 infection [3]. Infection in pregnancy, even if asymptomatic or mild, may have long-term impacts for a pregnant woman or impair the neurodevelopment of her child, a risk which is well established for viral infections such as Zika virus or cytomegalovirus [8].Data on the impact of SARS-CoV-2 infection in pregnancy on neurodevelopmental outcomes is emerging [9, 10]. Evidence regarding asymptomatic SARS-CoV-2 infection and adverse birth outcomes is limited and the long-term effects of the pandemic on infant health remain poorly understood [11].

Serological surveillance is a useful means of estimating population-level immunity against infectious diseases using cross-sectional studies of antibody prevalence [12]. In the case of SARS-CoV-2, serological surveys are helpful in estimating the number of people who have been exposed to SARS-CoV-2, whether they were symptomatic or not to better clarify the dynamics of exposure during the different epidemic waves as vaccines were rolled out [13]. For example, seroprevalence surveys conducted across Kenya, South Africa and Malawi have all reported community transmission, which is several times higher than that detected by national virological surveillance programmes [14–16].

Seroprevalence studies in pregnancy enables insights into the real magnitude of exposure to SARS-CoV-2 infections and the extent of under-reporting of SARS-CoV-2 cases. Information on seroprevalence in pregnancy, placental antibody transfer and antibody half-life also offer the possibility to approximate the number of mother-infant dyads who could potentially exhibit immunological protection against subsequent infections, especially with low vaccine coverage in many low-resource settings (LRS). Finally, seroprevalence studies can also provide

insight into the relationship between infection and vaccination, symptoms, and antibody responses to assist with future screening and prevention policies in pregnancy.

To address these specific gaps, we investigated the seroprevalence and the associations of different factors on seropositivity to the SARS-CoV-2 virus among pregnant women and their infants in Uganda and Malawi. This was performed during consecutive SARS-CoV-2 waves.

## Methods

### Study design and participants

PeriCOVID Africa is a multi-site prospective mother-infant cohort study using an adapted WHO UNITY protocol [17], whereby women were categorised into two categories dependent on serological testing using the Wantai total antibody assay as exposed (positive serology) or unexposed (negative serology) to SARS-CoV-2. Women were additionally screened for symptoms, using the WHO definition for probable COVID-19 disease at the time of study participation [18]. We defined symptomatic COVID-19 infection according to the WHO definitions of probable COVID-19 illness [18] and asymptomatic infection as seropositive or PCR positive at enrolment in the absence of reported symptoms. Unexposed women were those with no reported symptoms consistent with SARS-CoV-2 infection and negative serology.

**Recruitment.** Each study adapted the WHO UNITY protocol according to local needs and capacity considerations. Women were recruited into the study either during an antenatal visit, or during labour at seven study clinics and hospitals in Uganda and Malawi. In all studies, gestational age at enrolment was estimated by date of last menstrual period and fundal height. Additionally, in periCOVID Uganda and periCOVID Malawi Ballard scores were calculated at birth. Individual study recruitment, sampling and follow up are shown in Table 1. The first participant was recruited 1st February 2021. Final participant follow-up and sampling was concluded by 31st January 2022.

**COVID-19 testing.** Participating women at all study sites were screened for COVID-19 symptoms using a standardised data collection form with questions including a recent history of fever, cough, anosmia and ageusia and contact with a known SARS-Cov-2 case. Information on COVID-19 illness symptoms was collected at enrolment for the 14 days prior to enrolment in PeriCOVID Uganda and Malawi, and in the 28 days prior to enrolment in COMAC Uganda. Women enrolled in PeriCOVID Uganda had a nasal swab taken at enrolment to test for SARS-CoV-2 by PCR. In Malawi throat swabs were taken if the clinical syndrome was suggestive of a probable COVID-19 illness as defined by the WHO [18].

**Blood sampling.** Sampling at all sites for antibodies to SARS-COV-2 included at least a maternal venous and paired cord blood sample (see Table 1 for sampling schedule at each site).

**Data collection.** Each site (KNRH, Kawaala, Kitebi or Mukono General Hospital in Kampala, Uganda and QECH in Blantyre, Malawi) used a study questionnaire which was completed by research staff to capture information from study participants and then uploaded to a central RedCAP database. This included data on maternal age, significant past medical history, HIV and socioeconomic status; onset and duration of signs and symptoms of SARS-CoV-2 illness if present, and self-reporting of prior SARS-CoV-2 illness; gestational age at enrolment, parity, number of foetuses (if known before delivery), co-infection with malaria, vaccinations received in pregnancy including the SARS-CoV-2 vaccines; gestational age at delivery, delivery method, intrapartum and postpartum complications such as pre-term birth, stillbirth, abortion, and maternal death; neonatal outcomes including evidence of COVID-19 illness, Neonatal Intensive Care Unit (NICU) admission, low birth weight and neonatal death; infant health status.

**Table 1. Recruitment, sampling and follow-up by study site.**

| Study Site | Enrolment timing | Study period | Inclusion Criteria | Exclusion Criteria | Sampling at enrolment | Follow up |
|---|---|---|---|---|---|---|
| All sites | | | Willingness to provide informed consent<br>Pregnant women | As per individual study sites | Maternal and Cord blood | 6 week follow up |
| periCOVID Uganda | Antenatal clinic or delivery | February 2021 – January 2022 | Pregnant women (including emancipated minors aged over 14 years) at any gestation including the day of delivery<br>Planning to deliver at one of the designated study sites and willing to stay in the area for the first six weeks of their baby's life<br>Willing to attend a follow up visit at six weeks postpartum | No exclusion criteria | Maternal nasopharyngeal swab* | **For COVID-19 Cases only\*.**<br>**Maternal:** nasopharyngeal maternal blood sample (5ml serum), **Infant**: nasal swab, blood sample (2-5ml) or dried blood spot, |
| COMAC | Delivery | August 2021 –January 2022 | low risk of infection with tuberculosis in the household<br>mother of legal age (including if emancipated minor) for participation<br>mother residing within the study area, not intending to move out of the area in the next 4 months and is likely to be traceable for up to 12 months.<br>HIV-1 positive women should be receiving the necessary antiretroviral treatment and prophylaxis (Ugandan Option B + guidelines)<br>HIV-exposed babies received peri exposure prophylaxis (Ugandan Option B+ guidelines) | • Baby weighs less than 2kg at birth<br>• Baby requires hospital admission for severe illness at birth<br>• Serious congenital malformation(s)<br>• severely ill mother on the day of giving birth whose condition(s) require(s) hospitalization | Nil additional | **Maternal** blood test at 14 weeks. Nasopharyngeal swab at 6 weeks and 14 weeks. **Infant**: blood test at 14 weeks. Nasopharyngeal swab at 6 weeks and 14 weeks |
| Malawi | Delivery | March 2021 –January 2022 | All women presenting to QECH in labour with an estimated gestation of 28 weeks or greater who<br>Are willing to attend a follow up visit at 6 weeks postpartum | No exclusion criteria. | Nil additional | **For COVID-19 cases only.**<br>M**aternal** *blood, rectal swab, breastmilk. I**nfant\*:** blood |

## Laboratory methods

As per FIND guidelines at the time of the study protocol development [19], we used two different SARS-CoV-2 specific antibody assays that targeted either the receptor binding domain (RBD; total antibody), the spike protein (anti-S, total antibody) or nuclear capsid (anti-NCP, IgG). We performed in-house specificity and sensitivity testing, respectively, using 100 pre-COVID19 (pre-2019) samples selected by month for seasonality assessment and 20 PCR positive samples [19] to perform assay validation. We also examined potential cross-reactivity in our assay from malaria-specific antibodies using 74 women who tested positive to malaria (antibody positive by rapid diagnostic test (RDT) from pre-COVID samples and 15 SARS-COV-2 PCR positive samples in women who did not have malaria (negative RDT) during the COVID-19 pandemic. Results can be seen in S1 Table.

Laboratory testing for SARS-COV-2 antibodies was performed at the MRC/UVRI and LSHTM Uganda Research Unit or the Malawi Liverpool Wellcome facilities using the Wantai SARS-CoV-2 total antibody ELISA kit (Beijing Wantai Biological Pharmacy Enterprise Co., Ltd, Beijing, China). The manufacturer-reported assay sensitivity is 94.4%, with a specificity of

100%. All specimens that tested positive for Wantai were tested using Euroimmun Anti-SARS-CoV-2 NCP/S ELISA (IgG) (Euroimmun, Lübeck, Germany) kits for the detection of IgG antibodies to SARS-COV-2 nucleocapsid and spike (S1–S7 Tables) proteins, respectively. Euroimmun Anti-SARS-CoV-2 NCP/S ELISA (IgG) is a semi-quantitative immunoassay with a reported sensitivity of 94.6% and specificity of 99.8% in samples collected at least 10 days after confirmed SARS-COV-2 infection. A sample was considered positive if the Wantai test was positive. Results which were reported as borderline on the Euroinmmune assay were considered as negative for the purpose of our analysis. Due to variable specificity of the Euroimmune assay, we report Wantai results for all outcomes as per manufacturer's instructions.

As the Wantai ELISA is a qualitative test, WHO standards for NCP and S proteins were run on all assays. The stock concentration for NCP and S proteins was 123μg/ml and 1000μg/ml, respectively. The working concentration for both NCP and S proteins was 2 μg/ml. The calibration curve was created using WHO International standard for anti-SARS-CoV-2 immunoglobulin (NIBSC 20/136) using a 12-well dilution series created in 1.75-fold steps, starting at 1:200 and this series was used to generate the curve. All laboratory testing in Uganda for SARS-CoV-2 antibodies was performed using the ETI-MAX 3000 (Diasorin, Saluggia, Italy).

**Sample size calculations.** Although no sample size was possible at the time of study set up (within 6 months of the pandemic emerging), we estimated the standard error around seroprevalence based on the limited published data as follows. Given that the standard error is greatest (and therefore confidence intervals are widest) around a seroprevalence estimate of 50%, the maximum margin of error (half width of 95% confidence interval) was expected to be 1.8% for the largest country site (Uganda) and 4.9% for the smallest country site (Malawi) (assuming cord blood samples were obtained from all women delivering). Separate analyses for waves within individual countries would increase the half-width of the confidence interval to between 3.1 and 11.0%.

## Data analysis

The sero-epidemiological analyses were carried out using all participants with enrolment blood sample results available for analysis. The proportion of seropositive results was calculated for the individual waves of SARS-CoV-2 within each country. This was done for maternal blood samples to estimate the seroprevalence of SARS-CoV-2 among pregnant women, and for cord blood samples to estimate the seroprevalence of SARS-CoV-2 antibodies in infants. The dates used to define the SARS-COV-2 waves in each country are given in S2 Table and were taken from Our World in Data [20]. Waves were defined by taking the nadir between each peak for the start and end dates of each wave. Confidence intervals for prevalence estimates were computed using the Clopper-Pearson (Binomial Exact) method.

The geometric mean concentration (GMC) and 95% confidence intervals (CI) of anti-S and anti-NCP antibodies measured on the Euroimmune assay were calculated for mother-infant pairs. To study the rate of placental transfer of SARS-CoV-2 antibodies, the geometric mean ratio (GMR) of infant to maternal antibodies was calculated.

The proportion of participants with a symptomatic or asymptomatic infection was calculated for women who were seropositive at enrolment and for those who had a positive RT-PCR test at enrolment.

The impact of infection on key pregnancy and neonatal outcomes was modelled for women and infants in PeriCOVID Malawi and Uganda log-binomial generalised linear models (GLM) which were adjusted for country. Models were constructed for maternal death, infant death, combined adverse pregnancy outcome (at least one of maternal death, abortion, premature labour or stillbirth), and the combined adverse neonatal outcome (at least one of neonatal/

infant death, prematurity, low birth weight, NICU admission after birth, or birth asphyxia). Relative risks were presented with 95% confidence intervals for all models reaching convergence. Results were not presented for adverse outcomes with fewer than 5 events.

Statistical analyses were carried out using R version 4.2.1. No significance tests were conducted.

### Ethical considerations

The study documents were reviewed and approved by the Ethics Committee of the relevant institutions: Uganda: Makarere University School of Medicine (SOMREC), Uganda National Council for Science and Technology (UNCST); Regional Committees for Medical and Research Ethics in Norway; Malawi: College of Medicine Research Ethics Committee (COMREC). Informed consent was obtained from all participants. Patient information leaflets were available in English and in the local language for participants to read in their own time prior to consenting. Eligible participants who were illiterate were read the patient information sheet by a member of the research team with an independent witness present to verify the participant's understanding of the information.

## Results

In total, 1379 women were enrolled, giving birth to 1387 infants (Fig 1 and Table 2). The mean (SD) age of all women was 26 years (6) across the three sites. Most (n = 1346, 98%) pregnancies were singleton. The HIV prevalence was 9% (n = 272). 371 women delivered outside of a study hospital and so no blood samples were available for analysis. Deliveries of 1024 infants with cord blood samples occurred at study sites. Almost all were livebirths (n = 1009/1024, 99%) and most (n = 888/1024, 87%) were not admitted to the NICU after birth (Table 2). A total of 909/1379 (65.9%) women in periCOVID Uganda and periCOVID Malawi had a PCR result at enrolment available for analysis of which 68/909 (7.5%) women had positive PCR results for SARS-CoV-2. Amongst these women, 77.9% (n = 53) had symptoms consistent with COVID-19 disease. The majority (88.7%, n = 47/53) were in Malawi who were performing RT-PCR testing only on symptomatic women. In periCOVID Uganda, where all women had a RT-PCR test at enrolment, 31.6% (n = 6/19) of those with a positive RT-PCR test had symptoms suggestive of COVID-19 disease (S3 Table). Genotyping revealed all positive cases from Uganda to be of the delta variant.

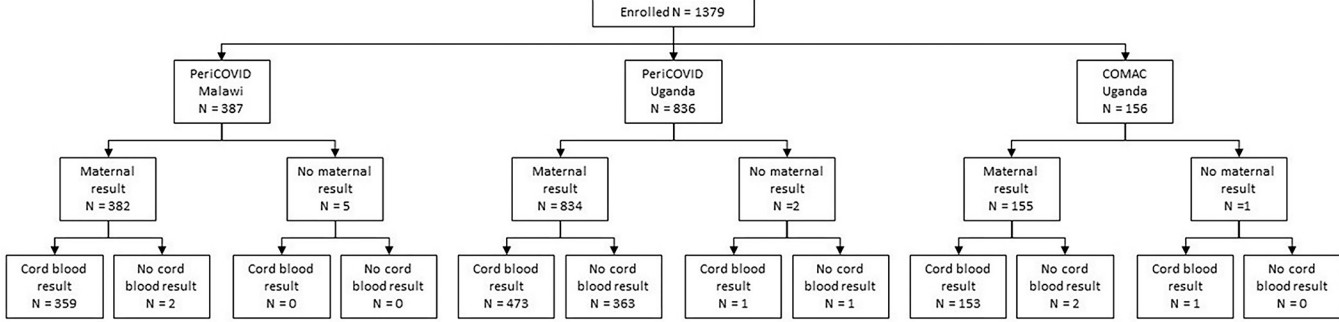

**Fig 1. Study flow chart.** Study flow chart to show the number of women enrolled by study, with maternal serology available at enrolment (maternal result) and the infants with cord blood serology (cord blood result) available.

**Table 2. Demographics for women and infants in the study.**

| Characteristic of women enrolled | Overall, N = 1,379 | PeriCOVID Malawi, N = 387 | PeriCOVID Uganda, N = 836 | COMAC Uganda, N = 156 |
|---|---|---|---|---|
| Age (years) Mean (SD) | 26 (6) | 25 (7) | 26 (6) | 27 (5) |
| Missing | 2 | 2 | 0 | 0 |
| Number of fetuses | | | | |
| Singleton | 1,346 (98%) | 371 (96%) | 819 (98%) | 156 (100%) |
| Twins | 30 (2.2%) | 14 (3.6%) | 16 (1.9%) | 0 (0%) |
| Triplets | 1 (<0.1%) | 0 (0%) | 1 (0.1%) | 0 (0%) |
| Missing | 2 | 2 | 0 | 0 |
| Highest level of education | | | | |
| Graduate Education /Terminal Degree Completed. | 23 (1.7%) | 1 (0.3%) | 20 (2.4%) | 2 (1.3%) |
| No Formal Education | 12 (0.9%) | 5 (1.3%) | 5 (0.6%) | 2 (1.3%) |
| Other | 2 (0.1%) | 0 (0%) | 2 (0.2%) | 0 (0%) |
| Primary Education Completed | 211 (15%) | 98 (26%) | 84 (10%) | 29 (19%) |
| Secondary Education Completed | 252 (18%) | 129 (34%) | 105 (13%) | 18 (12%) |
| Some Primary Education | 239 (17%) | 126 (33%) | 86 (10%) | 27 (17%) |
| Some Secondary Education | 509 (37%) | 0 (0%) | 432 (52%) | 77 (49%) |
| University / College Completed | 127 (9.2%) | 25 (6.5%) | 101 (12%) | 1 (0.6%) |
| Missing | 4 | 3 | 1 | 0 |
| HIV | | | | |
| No | 1,107 (91%) | 325 (84%) | 782 (94%) | 0 (NA%) |
| Yes | 114 (9.3%) | 60 (16%) | 54 (6.5%) | 0 (NA%) |
| Missing | 158 | 2 | 0 | 156 |
| **Characteristics of infants born and providing a cord blood sample** | **Overall, N = 1,024** | **PeriCOVID Malawi, N = 365** | **PeriCOVID Uganda, N = 503** | **COMAC Uganda, N = 156** |
| Status of infant at birth | | | | |
| Livebirth | 1,009 (99%) | 357 (99%) | 496 (99%) | 156 (100%)* |
| Miscarriage | 2 (0.2%) | 1 (0.3%) | 1 (0.2%) | 0 (0%) |
| Stillbirth | 10 (1.0%) | 4 (1.1%) | 6 (1.2%) | 0 (0%) |
| Missing | 3 | 3 | 0 | 0 |
| Gestation at birth (weeks) | 27, 44 | 27, 44 | 28, 44 | Inf, -Inf |
| Missing | 157 | 1 | 0 | 156 |
| Sex | | | | |
| Female | 492 (48%) | 164 (45%) | 248 (49%) | 80 (51%) |
| Male | 531 (52%) | 200 (55%) | 255 (51%) | 76 (49%) |
| Missing | 1 | 1 | 0 | 0 |
| Birth weight (grams) | | | | |
| Mean (SD) | 3,082 (540) | 2,910 (573) | 3,151 (504) | 3,262 (458) |
| Range | 600, 4,850 | 600, 4,500 | 1,200, 4,850 | 2,100, 4,700 |
| Missing | 8 | 2 | 6 | 0 |
| Admitted to NICU after birth | | | | |
| No | 888 (87%) | 264 (73%) | 468 (93%) | 156 (100%)* |
| Yes | 133 (13%) | 98 (27%) | 35 (7.0%) | 0 (0%) |
| Missing | 3 | 3 | 0 | 0 |

* COMAC Uganda only recruited infants who were born alive and not admitted to the NICU after birth

**Table 3. Maternal results.**

| Characteristic | PeriCOVID Malawi | | | PeriCOVID Uganda | | COMAC Uganda | |
|---|---|---|---|---|---|---|---|
| | Wave 2, N = 51 N (%) (95% CI) | Wave 3, N = 229 N (%) (95% CI) | Wave 4, N = 107 N (%) (95% CI) | Wave 1, N = 194 N (%) (95% CI) | Wave 2, N = 642 N (%) (95% CI) | Wave 2, N = 82 N (%) (95% CI) | Wave 3, N = 74 N (%) (95% CI) |
| Maternal blood sample | | | | | | | |
| Negative | 24 (51%) (36%, 66%) | 75 (33%) (27%, 39%) | 26 (24%) (17%, 34%) | 102 (53%) (45%, 60%) | 240 (38%) (34%, 41%) | 22 (27%) (18%, 38%) | 7 (9.6%) (3.9%, 19%) |
| Positive | 23 (49%) (34%, 64%) | 153 (67%) (61%, 73%) | 81 (76%) (66%, 83%) | 92 (47%) (40%, 55%) | 400 (62%) (59%, 66%) | 60 (73%) (62%, 82%) | 66 (90%) (81%, 96%) |
| Missing | 4 | 1 | 0 | 0 | 2 | 0 | 1 |
| Symptoms in those seropositive | | | | | | | |
| Asymptomatic | 21 (91%) (72%, 99%) | 134 (88%) (81%, 92%) | 57 (70%) (59%, 80%) | 92 (100%) (96%, 100%) | 362 (91%) (87%, 93%) | 60 (100%) (94%, 100%) | 65 (98%) (92%, 100%) |
| Symptomatic | 2 (8.7%) (1.1%, 28%) | 19 (12%) (7.6%, 19%) | 24 (30%) (20%, 41%) | 0 (0%) (0.00%, 3.9%) | 38 (9.5%) (6.8%, 13%) | 0 (0%) (0.00%, 6.0%) | 1 (1.5%) (0.04%, 8.2%) |
| Symptoms in those seronegative | | | | | | | |
| Asymptomatic | 24 (100%) (86%, 100%) | 61 (81%) (71%, 89%) | 12 (46%) (27%, 67%) | 101 (99%) (95%, 100%) | 223 (93%) (89%, 96%) | 22 (100%) (85%, 100%) | 7 (100%) (59%, 100%) |
| Symptomatic | 0 (0%) (0.00%, 14%) | 14 (19%) (11%, 29%) | 14 (54%) (33%, 73%) | 1 (1.0%) (0.02%, 5.3%) | 17 (7.1%) (4.2%, 11%) | | |

## Seropositivity in pregnant women and their infants

Overall, 1371/1379 maternal samples (382 from Malawi and 989 from Uganda) and 987/1024 cord blood samples (359 from Malawi and 628 from Uganda) were available for analysis (Tables 3 and 4). There were 875 SARS-CoV-2 seropositive women in the study (257 (72%) in Malawi, 618 (62%) in Uganda), of whom 791 (90.4%) were asymptomatic in the 14 days

**Table 4. Cord blood serology.**

| Characteristic | PeriCOVID Malawi | | | PeriCOVID Uganda | | COMAC Uganda | |
|---|---|---|---|---|---|---|---|
| | Wave 2, N = 52 N (%) (95% CI) | Wave 3, N = 230 N (%) (95% CI) | Wave 4, N = 111 N (%) (95% CI) | Wave 1, N = 195 N (%) (95% CI) | Wave 2, N = 642 N (%) (95% CI) | Wave 2, N = 82 N (%) (95% CI) | Wave 3, N = 74 N (%) (95% CI) |
| Concordance of maternal and cord blood samples | | | | | | | |
| Concordant | 35 (80%) (65%, 90%) | 194 (92%) (88%, 96%) | 101 (96%) (91%, 99%) | 79 (82%) (73%, 89%) | 271 (72%) (67%, 76%) | 81 (99%) (93%, 100%) | 68 (96%) (88%, 99%) |
| Discordant | 9 (20%) (9.8%, 35%) | 16 (7.6%) (4.4%, 12%) | 4 (3.8%) (1.0%, 9.5%) | 17 (18%) (11%, 27%) | 106 (28%) (24%, 33%) | 1 (1.2%) (0.03%, 6.6%) | 3 (4.2%) (0.88%, 12%) |
| Missing | 8 | 20 | 6 | 99 | 265 | 0 | 3 |
| Cord blood results with positive maternal blood result | | | | | | | |
| Negative | 8 (40%) (19%, 64%) | 14 (10%) (5.7%, 17%) | 3 (3.8%) (0.78%, 11%) | 8 (18%) (8.2%, 33%) | 38 (16%) (12%, 21%) | 0 (0%) (0.00%, 6.0%) | 3 (4.7%) (0.98%, 13%) |
| Positive | 12 (60%) (36%, 81%) | 123 (90%) (83%, 94%) | 77 (96%) (89%, 99%) | 36 (82%) (67%, 92%) | 198 (84%) (79%, 88%) | 60 (100%) (94%, 100%) | 61 (95%) (87%, 99%) |
| Missing | 3 | 15 | 3 | 48 | 165 | 0 | 2 |
| Cord blood result with negative maternal blood result | | | | | | | |
| Negative | 23 (96%) (79%, 100%) | 71 (97%) (90%, 100%) | 24 (96%) (80%, 100%) | 43 (83%) (70%, 92%) | 73 (52%) (43%, 60%) | 21 (95%) (77%, 100%) | 7 (100%) (59%, 100%) |
| Positive | 1 (4.2%) (0.11%, 21%) | 2 (2.7%) (0.33%, 9.5%) | 1 (4.0%) (0.10%, 20%) | 8 (15%) (6.9%, 28%) | 68 (48%) (40%, 57%) | 1 (4.5%) (0.12%, 23%) | 0 (0%) (0.00%, 41%) |
| Missing | 1 | 4 | 3 | 52* | 98 | | |

* 1 cord blood sample in PeriCOVID Uganda was insufficient for analysis

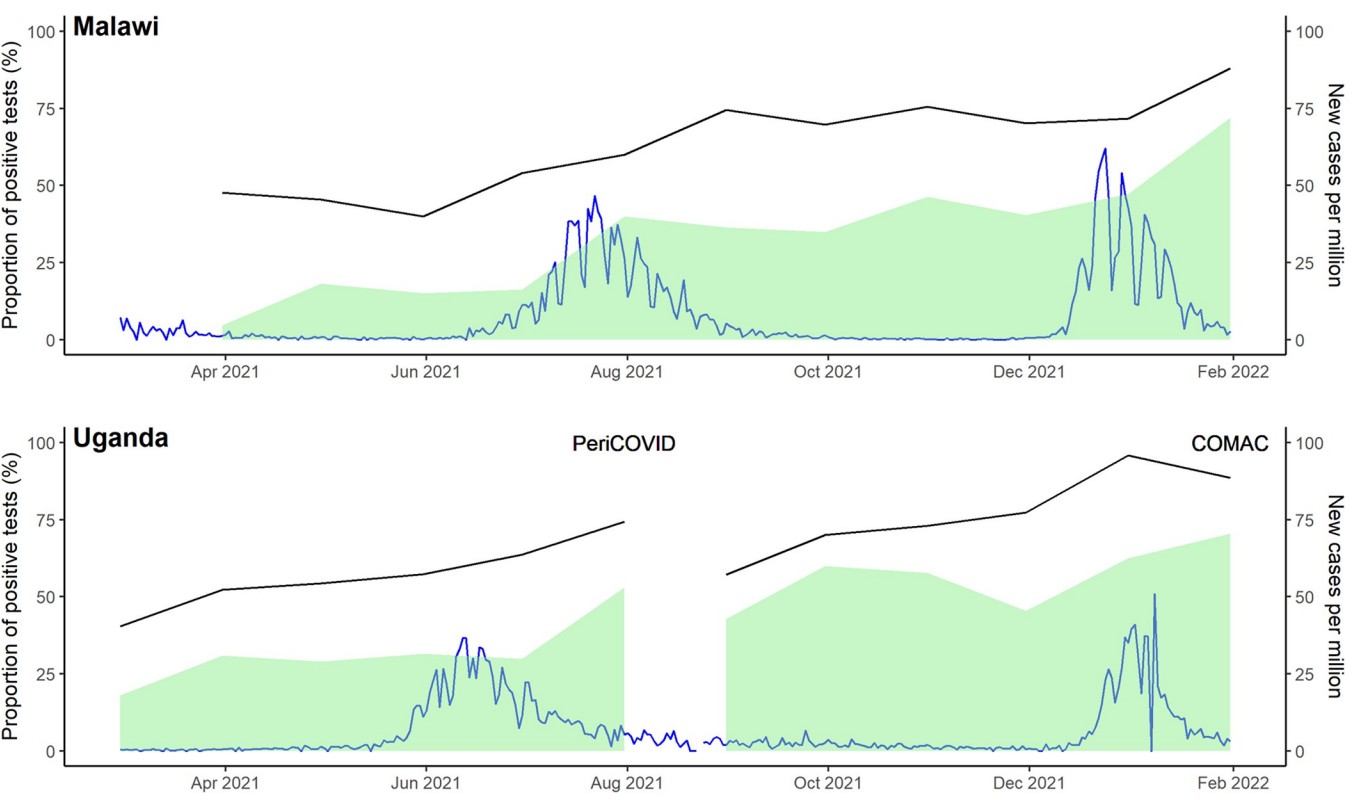

**Fig 2. Monthly sero-positivity by country in pregnant women.** Black line is the monthly proportion of results that were positive. Shaded in green is the proportion of Wantai positive samples that were also positive for anti-spike IgG in the Euroimmune assay. Blue line is the number of new cases per million in Malawi and Uganda, taken from Our World in Data.

(periCOVID Uganda and Malawi) -28 days (COMAC) prior to enrolment. This corresponded in Malawi to 21 asymptomatic seropositive participants during the second wave, 134 during the third wave and 57 in the fourth wave. In Uganda, this corresponded to 92 in the first wave, 422 in the second wave and 65 in the third wave (Table 3).

There was an increase in seropositivity with each subsequent wave, increasing from 49% (23/47) in the second wave to 76% (81/107) in the fourth wave in Malawi, and from 47% (92/194) in the first wave to 90% (66/73) in the third wave in Uganda. The majority of women with positive serology were asymptomatic in the 14 days prior to sampling, ranging from 70% (57/81 in the fourth wave in Malawi) to 100% (92/92 in the first wave in Uganda) (Table 3). Fig 2 shows the monthly total antibody positivity rate for each site with the daily number of new cases per million in Uganda and Malawi.

Table 4 shows high concordance between maternal and cord samples, ranging from 72% (271/377) to 99% (81/82).

## SARS-CoV-2 infection in pregnancy and key adverse pregnancy and neonatal outcomes

1220/1224 mothers had serology results at enrolment available for analysis and were included in the analysis of Sars-CoV-2 infection and pregnancy outcomes. 79/1220 mothers experienced at least one adverse pregnancy outcome (maternal death N = 4, abortion N = 4, premature labour N = 52, and stillbirth N = 26) and there were 46 infant deaths (S4 and S5 Tables). Of the 79 adverse pregnancy outcomes, 34 (43%) mothers were sero-negative, and 45 (57%)

**Table 5. Impact of infection (seropositivity) on key pregnancy and neonatal outcomes in periCOVID Malawi and periCOVID Uganda.**

| | Number included in model | Number of events | Relative Risk | 95% Confidence Interval |
|---|---|---|---|---|
| **Maternal death** | | | | |
| Sero-Negative | | 1 | — | — |
| Sero-Positive | | 3 | — | — |
| **Infant death** | | | | |
| Sero-Negative | 471 | 20 | — | — |
| Sero-Positive | 752 | 26 | 0.81 | 0.46, 1.46 |
| **Premature labour** | | | | |
| Sero-Negative | 476 | 20 | — | — |
| Sero-Positive | 758 | 32 | 0.97 | 0.57, 1.71 |
| **Still birth** | | | | |
| Sero-Negative | 473 | 15 | — | — |
| Sero-Positive | 752 | 11 | 0.48 | 0.22, 1.03 |
| **Abortion** | | | | |
| Sero-Negative | | 0 | — | — |
| Sero-Positive | | 4 | — | — |
| **Combined adverse pregnancy outcome** | | | | |
| Sero-Negative | 463 | 34 | — | — |
| Sero-Positive | 742 | 45 | 0.81 | 0.53, 1.26 |
| **Low birth weight** | | | | |
| Sero-Negative | 474 | 22 | — | — |
| Sero-Positive | 753 | 39 | 0.98 | 0.60, 1.65 |
| **NICU admission** | | | | |
| Sero-Negative | 471 | 63 | — | — |
| Sero-Positive | 753 | 107 | 0.96 | 0.73, 1.28 |
| **Combined adverse neonatal outcome** | | | | |
| Sero-Negative | 476 | 77 | — | — |
| Sero-Positive | 758 | 120 | 0.92 | 0.71, 1.19 |

Models are adjusted for country

mothers were sero-positive, compared to 435 (38%) mothers who were sero-negative and 706 (62%) mothers who were sero-positive with no adverse pregnancy outcomes. There was no difference in pregnancy outcomes due to sero-positive SARS-CoV-2 status, as shown by relative risks (95% confidence intervals) in the range of 0.48 (0.22, 1.03) to 0.98 (0.60, 1.65) (Table 5). Due to the small numbers of outcomes (N = 4), the impact of Sars-CoV-2 infection on maternal death could not be modelled. There was also no difference in outcomes due to SARS-CoV-2 infection with and without symptoms in the 14 days (periCOVID Uganda and Malawi) -28 days (COMAC) prior to enrolment, as shown by relative risks (95% confidence intervals) in the range of 0.48 (0.21, 1.05) to 1.39 (0.64, 2.68) (S6 Table).

There were 197 adverse infant outcomes (at least one of: infant/neonatal death, prematurity, low birth weight, NICU admission and birth asphyxia). There were 46 infant deaths, 26 (57%) from sero-positive and 20 (43%) from sero-negative women (S5 Table). There was no difference in risk of infant death due to SARS-CoV-2 serology status in the mother (Table 5). 77 (39%) infants with adverse outcomes were born to women who were sero-negative and 120 (61%) sero-positive. The relative risk was 0.92 (95% CI 0.71, 1.19), providing no evidence of a difference in the risk of at least one adverse neonatal outcome due to serological status of the

mother. There was also no evidence of a difference in neonatal outcome due to positive sero-logical status with and without symptoms in the mother (S4 and S5 Tables).

## Placental transfer of SARS-CoV-2 antibodies in those with prior infection and/or vaccination

In PeriCOVID Uganda, 208/503 mother-infant pairs had anti-S IgG results available for analysis, corresponding to 27 and 181 during the first and second waves respectively. There was no difference between the maternal and cord blood anti-S IgG GMCs during the first wave. The GMR (95% confidence interval) in the second wave was 1.7 (1.3, 2.3), indicating that anti-S IgG was higher in the cord blood than the maternal blood at enrolment. In comparison, in COMAC Uganda the GMR (95% CI) was 1.6 (0.8, 3) and 0.7 (0.4, 1) for mother-infant pairs enrolled during the second and third waves respectively, indicating no difference between the maternal and cord blood anti-S IgG for the 60 mother-infant pairs enrolled during the second wave, or for the 59 enrolled during the third wave (S7 Table). The rate of placental transfer of anti-S IgG is plotted in Fig 3.

There were 39 mother-infant pairs enrolled in PeriCOVID Uganda during the first wave with anti-NCP results, and 194 during the second wave. For both waves, there was no evidence of a difference in the maternal and cord blood anti-NCP IgG results, as shown by GMRs of 1 (0.6, 1.6) and 0.9 (0.7, 1.3) respectively. In COMAC Uganda, 55 and 43 mother-infant pairs enrolled during the second and third waves, respectively, had anti-NCP IgG results. The GMR

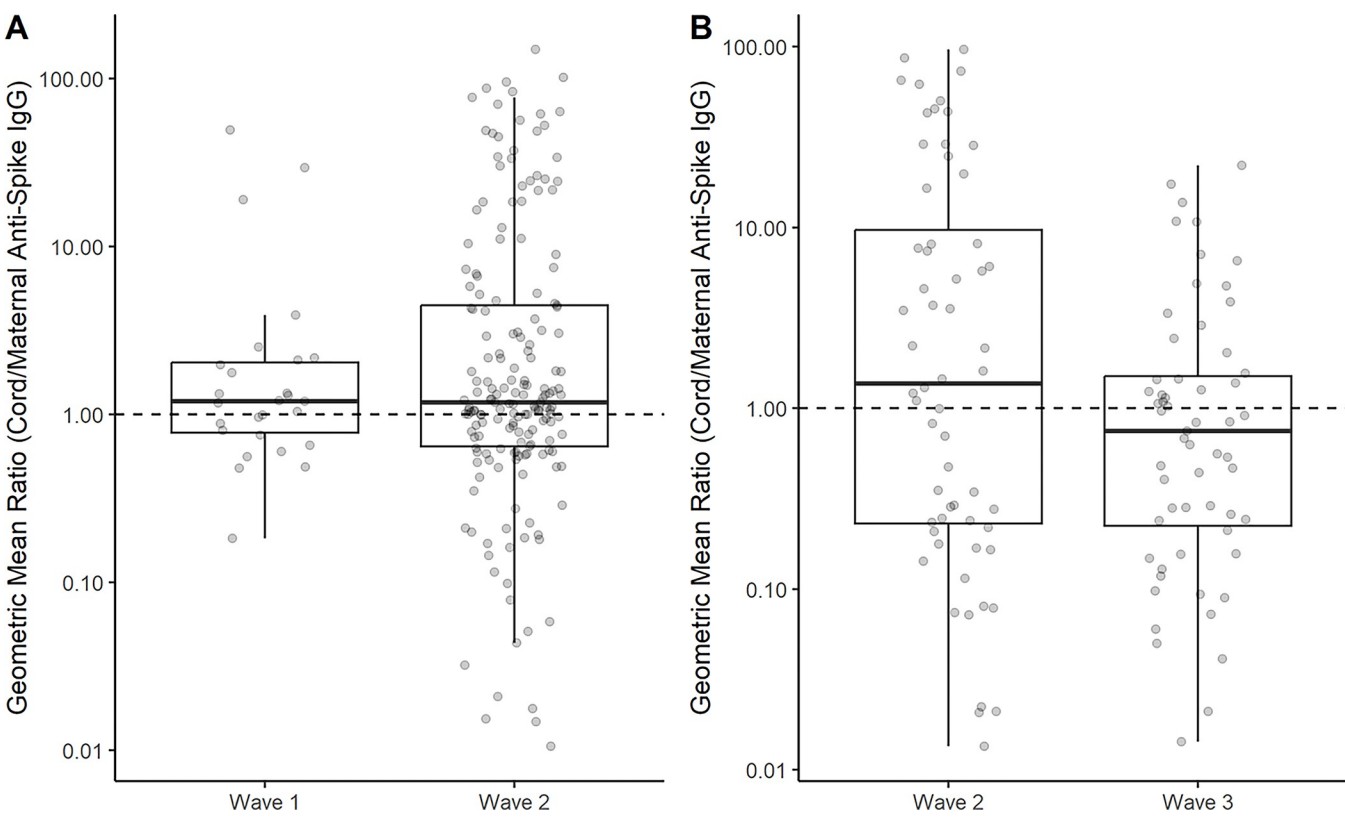

**Fig 3.** Placental transfer of anti-s IgG in A) PeriCOVID Uganda and B) COMAC Uganda. Geometric mean ratios (GMRs) of anti-spike IgG for each mother-infant pair with results available for analysis on the Euroimmune anti-spike IgG assay. Boxplots of the GMRs show no evidence of a difference in placental transfer in different waves of the pandemic.

in the second wave was 1.8 (1.1, 2.9), and in the third wave was 2.7 (1.7, 4.4), indicating that anti-NCP IgG was higher in cord blood samples than in the maternal blood (S7 Table). The rate of placental transfer of anti-NCP IgG is plotted in S1 Fig. There was no clear evidence of a difference in placental transfer of both anti-S and anti-NCP IgG during different waves (Fig 3 and S1 Fig).

## Number of vaccinated women during pregnancy

A total of 29 participants across all sites reported prior vaccination to SARS-CoV-2: 7 from Malawi, 1 from periCOVID Uganda and 21 from COMAC. Monthly numbers of positive results in COMAC Uganda by vaccination status can be seen in S2 Fig.

## Discussion

This study describes the increasing prevalence of SARS CoV-2 infection across 2 countries and 5 hospital sites in East and southern Africa, and across several COVID-19 waves. This increase in prevalence coincided with waves of delta and omicron infection within the countries, respectively. Studies included in a systematic review and meta-analysis of anti-SARS-CoV-2 seroprevalence in Africa showed wide-variation between countries (seroprevalence estimates ranged from 0% to 63%) [21]. A study of antenatal care clinics in two Kenyan referral hospitals found highest seroprevalence of up to 85% in Busia [22]. Our data indicates that the majority of cases were asymptomatic and adds to existing evidence that suggests under-reporting of infection if based solely on confirmed cases by PCR [4, 5, 21].

The high prevalence of poor maternal and child health outcomes in sub-Saharan Africa, combined with the known impact of SARS-CoV-2 illness in pregnancy from existing studies outside of Africa [23–27], means that we need to better understand the direct effects of exposure to COVID-19 in pregnancy and outcome for the pregnant woman and her infant. The INTERCOVID study [28], which included 2 sites in West Africa (Nigeria and Ghana) showed that infection in pregnancy was associated with increased maternal and neonatal morbidity and mortality. The AFREHEALTH study of 1315 hospitalized pregnant and non-pregnant women with and without SARS-CoV-2 revealed an increased risk of ICU admission and in-hospital death amongst pregnant women with COVID-19 [29]. Though we are unable to assess outcomes for symptomatic SARS-CoV-2 illness in our cohort due to low numbers our study does highlight that asymptomatic infection does not appear to be associated with death in the mother, or with worse neonatal outcomes in the first month of life. This is reassuring to parents and health care providers.

Furthermore, in our study placental transfer of IgG increased during each subsequent wave. Previous studies earlier in the pandemic suggested reduced placental transfer of IgG in women with a positive SARS-CoV-2 RT-PCR [30, 31]. More research is needed to better understand placental transfer with different SARS-CoV-2 variants.

### Limitations

Our study is limited by differences in methodology across sites, with sampling performed at different time points in pregnancy, despite efforts to adhere to the UNITY protocol. Though we were able to collect nose and throat swabs for SARS-CoV-2 RT-PCR in periCOVID Uganda and Malawi we were unable to do so in the COMAC study. Furthermore, in Malawi only symptomatic women were screened, which may skew the RT-PCR results. The uniform collection of cord blood across all sites however enables comparison across sites and strengthens our results. The ability to rapidly incorporate detection of a novel infection within existing cohort studies highlights the research capacity within study sites in low-resource settings. We

also note that study sites in both countries were in urban centres. Extrapolating data to rural communities or to other low-resource settings is not feasible.

We note that 27.2% (368/1355) of cord blood samples were not available for analysis, with majority of missing data occurring in periCOVID Uganda (n = 364). In PeriCOVID Uganda, a strict period of lockdown over the summer of 2021 with a ban on public transport made it challenging for participants to attend hospital for delivery, leading to a lower number of cord blood samples than anticipated. Calls to participants by healthcare visitors were increased to ensure that women were aware that study staff were still working and could care for them during their delivery.

We report our primary outcomes using the Wantai assay, but for placental antibody transfer, we report IgG using Euroimmune results. We identified cross-reactivity of antibodies against *Plasmodium falciparum* or other common cold coronaviruses (CCCs) as has been reported elsewhere in East Africa [21], meaning these results should be reviewed with caution. We had initially planned to use the Euroimmune anti-NCP assay to differentiate between infection and vaccination. However, the specificity of the assay precluded its use for this purpose. The Euroimmune anti-NCP assay has a lower sensitivity than other assays [32]. Several studies have shown a low anti-NCP positivity after mild infections [33]. As the pandemic progressed the chance of repeat SARS-CoV-2 infection increased, though these infections were generally milder [33]. Anti-NCP antibodies in some studies have remained negative in individuals who were vaccinated against SARS-CoV-2 and who had an rt-PCR confirmed illness [33]. A lower anti-NCP seropositivity later in the pandemic may therefore represent assay sensitivity and a lower anti-NCP immune response following mild or asymptomatic infection.

## Conclusion

Data from Uganda and Malawi showed a seroprevalence of SARS-CoV-2 higher than the cases figures identified by other sources, with asymptomatic infection being common. In future pandemics and outbreaks, seroprevalence studies may be a more accurate measure in assessing the true prevalence of infection and may guide vaccination strategy in vulnerable groups.

## Supporting information

**S1 Fig.** Placental transfer of anti-n IgG in A) PeriCOVID Uganda and B) COMAC Uganda. Geometric mean ratios (GMRs) of anti-nucelocapsid IgG for each mother-infant pair with results available for analysis on the Euroimmune anti-nucleocapsid IgG assay. Boxplots of the GMRs show no evidence of a difference in placental transfer in different waves of the pandemic.
(TIF)

**S2 Fig. Monthly number of positives in COMAC Uganda, by vaccination status.** The monthly number of women in COMAC Uganda who were seropositive at enrolment on each of the Wantai (top panel), Euroimmune anti-S (middle panel) and Euroimmune anti-N (bottom panel) assays, coloured by vaccination status. Green shows those who were unvaccinated at enrolment, blue shows the small proportion who were vaccinated, and red shows those whose vaccination status was unknown at enrolment. The majority were unvaccinated.
(TIF)

**S1 Table. Wantai Assay specificity.** Specificity of Wantai assay when tested on pre-COVID samples with and without malaria.
(DOCX)

**S2 Table. Dates used to define COVID-19 waves in Malawi and Uganda.** Dates used to define the COVID-19 waves, taken from Our World in Data.
(DOCX)

**S3 Table. Maternal PCR tests by study site and wave.** Symptoms are those defined as probable infection according to WHO criteria [18].
(DOCX)

**S4 Table. Adverse pregnancy outcomes in mothers enrolled in periCOVID Malawi and periCOVID Uganda.**
(DOCX)

**S5 Table. Adverse neonatal outcomes in infants born in periCOVID Malawi and periCOVID Uganda.**
(DOCX)

**S6 Table. Impact of infection (seropositive and WHO probable) on key pregnancy and neonatal outcomes.**
(DOCX)

**S7 Table. Placental transfer of anti-S and anti-N in Uganda.**
(DOCX)

## Acknowledgments

The periCOVID Consortium consists of:

Gladys Gadama[1], Priscilla Precious Mvula-Mtila[1], Queen Dube[2], Charlotte Van der veer[3], Louise Afran[3], Maryke Joanne Nielsen[3], Zione Pondani[3], Smart Njowa[3], Catherine Anscombe[3], Esperança Sevene[4,5], Sónia Maculuve[4], Anifa Valá[4], Angela Koech[6], Geoffrey Omuse[7], Marleen Temmerman[6]

 <u>Malawi</u>

1. Department of Obstetrics, Queen Elizabeth Central Hospital, Kamuzu University of Health Sciences

2. Department of Paediatrics Queen Elizabeth Central Hospital, Kamuzu University of Health Sciences

3. Malawi-Liverpool-Wellcome Trust, University of Liverpool

 <u>Mozambique</u>

4. Manhiça Health Research Center, Manhiça, Mozambique

5. Department of Physiological Science, Faculty of Medicine, Eduardo Mondlane University, Maputo, Mozambique

 <u>Kenya</u>

6. Centre of Excellence in Women and Child Health, Aga Khan University, Kenya

7. Department of Pathology, Aga Khan University, Kenya

We would also like to thank all study and laboratory staff who enabled recruitment, follow-up, and study testing.

## Author Contributions

**Conceptualization:** Bridget Freyne, Kirsty Le Doare.

**Data curation:** Lauren Hookham, Liberty Cantrell, Victoria Nankabirwa.

**Formal analysis:** Lauren Hookham, Liberty Cantrell, Stephen Cose, Merryn Voysey, Kirsty Le Doare.

**Funding acquisition:** Bridget Freyne, Victoria Nankabirwa, Halvor Sommerfelt, Kirsty Le Doare.

**Investigation:** Lauren Hookham, Luis Gadama, Esther Imede, Kirsty Le Doare.

**Methodology:** Liberty Cantrell, Stephen Cose, Merryn Voysey, Kirsty Le Doare.

**Supervision:** Lauren Hookham, Stephen Cose, Bridget Freyne, Luis Gadama, Kondwani Kawaza, Samantha Lissauer, Phillipa Musoke, Victoria Nankabirwa, Musa Sekikubo, Halvor Sommerfelt, Merryn Voysey, Kirsty Le Doare.

**Writing – original draft:** Lauren Hookham, Liberty Cantrell, Esther Imede, Merryn Voysey, Kirsty Le Doare.

**Writing – review & editing:** Lauren Hookham, Liberty Cantrell, Stephen Cose, Bridget Freyne, Esther Imede, Victoria Nankabirwa, Halvor Sommerfelt, Merryn Voysey, Kirsty Le Doare.

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
