## [Decision Letter · Decision Letter 0]

11 Sep 2023

PONE-D-23-25991Seroepidemiology of COVID-19 in pregnant women and their infants in Uganda and Malawi across multiple waves 2020-2022PLOS ONE

Dear Dr. Hookham,

Thank you for submitting your manuscript to PLOS ONE. After careful consideration, we feel that it has merit but does not fully meet PLOS ONE’s publication criteria as it currently stands. Therefore, we invite you to submit a revised version of the manuscript that addresses the points raised during the review process.

ACADEMIC EDITOR: We consider your manuscript competent, and meets the requirements for publication on PLOS ONE, however, there are some very important reviews needed in the manuscript. Attend to the comments raised by reviewers and provide your answers accordingly. Regards.

We look forward to receiving your revised manuscript.

Kind regards,

Olatunji Matthew Kolawole, Ph.D.

Academic Editor

PLOS ONE

Journal Requirements:

"periCOVID Uganda: The European and Developing Countries Clinical Trials Partnership (EDCTP) (grant number RIA2020EF- 2926 periCOVID Africa)

COMAC: The Research Council of Norway (grants 312768 and 223269) and 

The European and Developing Countries Clinical Trials Partnership (EDCTP) (grant number RIA2020EF- 2926 periCOVID Africa)

periCOVID Malawi: The European and Developing Countries Clinical Trials Partnership (EDCTP) (grant number RIA2020EF- 2926 periCOVID Africa)"

6. We notice that your supplementary [S1-2 Fig and S1-2 Table] are included in the manuscript file. Please remove them and upload them with the file type 'Supporting Information'. Please ensure that each Supporting Information file has a legend listed in the manuscript after the references list.

Additional Editor Comments:

The reviewers have made some comments that require attention in order to completely make this manuscript fit for publication on PLOS ONE. Kindly attend to these comments accordingly.

Reviewers' comments:

Reviewer's Responses to Questions

**Comments to the Author**

1. Is the manuscript technically sound, and do the data support the conclusions?

Reviewer #1: Yes

Reviewer #2: Yes

Reviewer #3: Partly

2. Has the statistical analysis been performed appropriately and rigorously? 

Reviewer #1: Yes

Reviewer #2: I Don't Know

Reviewer #3: Yes

3. Have the authors made all data underlying the findings in their manuscript fully available?

Reviewer #1: Yes

Reviewer #2: Yes

Reviewer #3: Yes

4. Is the manuscript presented in an intelligible fashion and written in standard English?

Reviewer #1: Yes

Reviewer #2: Yes

Reviewer #3: Yes

5. Review Comments to the Author

Reviewer #1: Laboratory method (158) and Laboratory analysis (170) need to be synchronised as it seems confusing at this time. Both described Laboratory assays performed in the study.

194 Although there was no predefined sample size, we aimed to obtain samples...... the study is already completed hence these should not be futuristic but past tense. Also, there is need to justify why no predetermine sample size.

237 ... The mean 237 age of all women was 26 years (SD 6) ........ should be rephrased to The mean (SD) age of all women was 26 (6) years .....

271 ......and the majority (70-100%) of women..... The actual number should also be included

273 The quality of the all the figures too poor for publication purposes

274 .....maternal and cord samples was high (72-99%).... The actual number should also be included

345 The authors should present the and key adverse pregnancy and neonatal outcomes among

350 The discussion is too brief and did not cover the results showed in the manuscript.

Reviewer #2: The authors have presented valuable data that would increase our understanding of COVID-19 and especially in a very important population- pregnant women and neonates.

This manuscript requires some revisions.

Table1: Exclusion criteria should be clearly stated as well as the inclusion criteria

Line 133: The opening statement "All women" I think this should be consenting or participating women, except all the women attending the centres participated in this study.

Line228: Ethical considerations. It should be stated if and how informed consent was obtained for all participants. This should be indicated as part ofcriteria for exclusion or inclusion.

Table 2: Age should be 'Mean age', I think.

Line 291: this section talks about adverse pregnancy outcomes in relation to SARS-CoV-2 infection however, the only results presented here is from serology. How have authors ascertained that Abs measured in pregnant women in thisstudy were due to infection and not vaccination? Also, since the Ab measured is of the IgG class, it is not certain when the infection occured either previously before the pregnancy or during the pregnancy.

Reviewer #3: Just a few reviews and recommendations.

1. Your title is clear and informative, but it could be more concise. You could consider removing some details such as the study name, the number of waves, and the countries, and include them in the abstract or the introduction instead. You could remove some details from your title because it is very long and it may not fit the journal's requirements or the readers' attention span. A shorter and simpler title may be more appealing and memorable for your audience. The details that you could remove from your title are not essential for describing your main topic or objective, and they could be easily included in the abstract or the introduction, where you have more space and opportunity to explain them. For example, you could write a title like this: Seroepidemiology of COVID-19 in pregnancy and infancy in sub-Saharan Africa. And then you could mention the study name, the number of waves, and the countries in the first paragraph of your abstract or introduction, such as: This paper reports the results of the PeriCOVID Africa study, a South-South-North partnership involving hospitals and health centres in five countries: Malawi, Uganda, Mozambique, The Gambia, and Kenya. We describe the seroepidemiology of SARS-CoV-2 infection in pregnant women enrolled in sites in Uganda and Malawi across three waves of COVID-19 (alpha, delta, and omicron) between 2020 and 2022, and the impact of SARS-CoV-2 infection on pregnancy and infant outcomes. This way, you can still provide all the relevant information about your study, but in a more organized and concise way.

2. Your abstract is well-structured and covers the main aspects of your study, but it is too long for PLOS ONE standards. You should shorten some sentences and remove some unnecessary information. You should also include some key numbers or statements that reflect your main results and conclusions.

3. Your results section is concise and clear, but it could be more informative. You should include some more details or numbers to support your statements, such as the exact seroprevalence rates by site and wave, the confidence intervals for the GMR, and the p-values for the association between SARS-CoV-2 antibody positivity and adverse outcomes. You should also use tables or figures to display your data more effectively and visually.

4. Your discussion section is well-written and covers the main points of your study, but it could be more balanced and critical. You should acknowledge some of the potential sources of bias or confounding in your study, such as the selection of participants, the timing of sample collection, the accuracy of serological tests, and the measurement of outcomes. You should also compare and contrast your findings with other studies from similar or different settings, and explain why they may agree or disagree. You should also suggest some specific actions or recommendations based on your results, such as the need for vaccination, surveillance, or prevention strategies for pregnant women and infants in SSA.

5. You should check the formatting of your references, tables, and figures according to the journal’s guidelines. You should also proofread your paper for spelling, grammar, and punctuation errors. You should also ensure that you have followed the appropriate reporting guidelines for your study design, such as STROBE for observational studies. You should also make your data and code available in a public repository or explain why they are not. You should also declare any competing interests or funding sources. You should also obtain ethical approval and consent from the participants.

6. PLOS authors have the option to publish the peer review history of their article (what does this mean?). If published, this will include your full peer review and any attached files.

Reviewer #1: **Yes: **Dr. Olufemi Samuel AMOO

Reviewer #2: **Yes: **Olusola Anuoluwapo Akanbi

Reviewer #3: **Yes: **Oluwaseun Paul Amoo

---

## [Author Response · Author response to Decision Letter 0]

21 Nov 2023

Dear reviewers, 

We thank you for your helpful and considerate comments on this piece. We have addressed them individually, as per the table below. 

With thanks for your time and consideration, 

Dr. Lauren Hookham 

Reviewer Comment To respond

Reviewer #1: Laboratory method (158) and Laboratory analysis (170) need to be synchronised as it seems confusing at this time. Both described Laboratory assays performed in the study.

 WE have removed “laboratory analysis” from the manuscript. Line 157 – changed from three to two assays used. 

 194 Although there was no predefined sample size, we aimed to obtain samples...... the study is already completed hence these should not be futuristic but past tense. Also, there is need to justify why no predetermine sample size. Line 190 – We have updated the discussion on sample size. 

 237 ... The mean 237 age of all women was 26 years (SD 6) ........ should be rephrased to The mean (SD) age of all women was 26 (6) years ..... Line 247 – changed as per reviewer comment. 

 271 ......and the majority (70-100%) of women..... The actual number should also be included Line 305 – We have added the actual numbers

 273 The quality of the all the figures too poor for publication purposes We have improved the quality of all figures

 274 .....maternal and cord samples was high (72-99%).... The actual number should also be included Line 309 – We have added the actual numbers

345 The authors should present the and key adverse pregnancy and neonatal outcomes among Thank you – we note this comment has been curtailed. We are happy to address if further information given. 

We have added additional statistics (relative risk and confidence intervals) for pregnancy outcomes by COVID status (313-318). 

350 The discussion is too brief and did not cover the results showed in the manuscript.

 Thank you 

The discussion section reports the primary outcome of the study and highlights the finding of high seroprevalence (line 349-364). We have added additional detail on other seroprevalence studies undertaken within Africa (line 372 – 377) 

We review previous studies undertaken in Africa on the impact of symptomatic COVID-19 infection in pregnancy (line 369-379). We are unable to assess for outcomes of women with symptomatic SARS-CoV-2 illness in our cohort due to low numbers. We do highlight that asymptomatic infection does not appear to be associated with worse outcomes (377 – 381). 

We discuss the results of placental transfer studies (line 393 – 396), with reference to existing data. We suggest additional research is needed to understand placental transfer with different SARS-CoV-2 variants

In addition, we discuss our limitations and sources of bias (400 – 409). 

Reviewer 2 

Table1: Exclusion criteria should be clearly stated as well as the inclusion criteria Line 127 - Table 1 includes inclusion and exclusion criteria by study site as per their study protocols. 

Line 133: The opening statement "All women" I think this should be consenting or participating women, except all the women attending the centres participated in this study. Line 131 – changed from all women to participating women. 

Line228: Ethical considerations. It should be stated if and how informed consent was obtained for all participants. This should be indicated as part ofcriteria for exclusion or inclusion. Line 239 - 243 – additional information regarding consent process added to text. 

 Table 2: Age should be 'Mean age', I think. Table 2 – added mean age (SD) to table 

 Line 291: this section talks about adverse pregnancy outcomes in relation to SARS-CoV-2 infection however, the only results presented here is from serology. How have authors ascertained that Abs measured in pregnant women in this study were due to infection and not vaccination? Also, since the Ab measured is of the IgG class, it is not certain when the infection occured either previously before the pregnancy or during the pregnancy. Line 364-366– 

We are aware of a minority of participants (n-29) who were vaccinated against SARS-Cov-2 during this study. Most participants were enrolled, with samples taken, prior to wide-spread availability of vaccine within each country setting. 

IgG against SARS CoV-2 can persist for many months, and the reviewer is correct in stating that for women with positive serology at enrolment we are unable to ascertain exactly when infection occurred and in some cases may have occurred prior to pregnancy. However, given that we undertook sampling soon after the initial waves of the pandemic occurred in each country, we feel this is the best estimate of COVID19 exposure in the absence of vaccination. 

In periCOVID Uganda, there were 2 clinical sampling time-points. Enrolment and delivery. There was a slight increase in discordant results between waves in the 2 Uganda sites (see Table 4). 

Reviewer 3 1. Your title is clear and informative, but it could be more concise. You could consider removing some details such as the study name, the number of waves, and the countries, and include them in the abstract or the introduction instead. You could remove some details from your title because it is very long and it may not fit the journal's requirements or the readers' attention span. A shorter and simpler title may be more appealing and memorable for your audience. The details that you could remove from your title are not essential for describing your main topic or objective, and they could be easily included in the abstract or the introduction, where you have more space and opportunity to explain them. For example, you could write a title like this: Seroepidemiology of COVID-19 in pregnancy and infancy in sub-Saharan Africa. And then you could mention the study name, the number of waves, and the countries in the first paragraph of your abstract or introduction, such as: This paper reports the results of the PeriCOVID Africa study, a South-South-North partnership involving hospitals and health centres in five countries: Malawi, Uganda, Mozambique, The Gambia, and Kenya. We describe the seroepidemiology of SARS-CoV-2 infection in pregnant women enrolled in sites in Uganda and Malawi across three waves of COVID-19 (alpha, delta, and omicron) between 2020 and 2022, and the impact of SARS-CoV-2 infection on pregnancy and infant outcomes. This way, you can still provide all the relevant information about your study, but in a more organized and concise way. Thank you 

- We have removed the following words from the title “across multiple waves 2020-2022”, and added the term cohort to better clarify methodology in the title as per STROBE guidelines. 

 2. Your abstract is well-structured and covers the main aspects of your study, but it is too long for PLOS ONE standards. You should shorten some sentences and remove some unnecessary information. You should also include some key numbers or statements that reflect your main results and conclusions.

 Thank you. 

We have reviewed the abstract and it now falls within the word limit of less than 300 words. 

 3. Your results section is concise and clear, but it could be more informative. You should include some more details or numbers to support your statements, such as the exact seroprevalence rates by site and wave, the confidence intervals for the GMR, and the p-values for the association between SARS-CoV-2 antibody positivity and adverse outcomes. You should also use tables or figures to display your data more effectively and visually. Thank you.

We have updated the results section to include seroprevalence rates by site and wave, and double checked that all GMRs have confidence intervals and updated the text to include these as needed.

All analysis for the study was descriptive so we have not presented p-values. We have instead added the relative risks and 95% confidence intervals for the association between SARS-CoV-2 antibody positivity and adverse outcomes.

 4. Your discussion section is well-written and covers the main points of your study, but it could be more balanced and critical. You should acknowledge some of the potential sources of bias or confounding in your study, such as the selection of participants, the timing of sample collection, the accuracy of serological tests, and the measurement of outcomes. You should also compare and contrast your findings with other studies from similar or different settings, and explain why they may agree or disagree. You should also suggest some specific actions or recommendations based on your results, such as the need for vaccination, surveillance, or prevention strategies for pregnant women and infants in SSA. Thank you. 

We note potential sources of bias within our study, specifically differences in methodology across the different sites (line 388). We have added additional information and discussion on differences in methodology. We have discussed missing data (see line 399) in more detail. 

We discuss accuracy of serological testing from line 419-432. We have added action points in the conclusion, stressing the importance of seroprevalence studies and their use in guiding vaccination strategies. 

 5. You should check the formatting of your references, tables, and figures according to the journal’s guidelines. You should also proofread your paper for spelling, grammar, and punctuation errors. You should also ensure that you have followed the appropriate reporting guidelines for your study design, such as STROBE for observational studies. You should also make your data and code available in a public repository or explain why they are not. You should also declare any competing interests or funding sources. You should also obtain ethical approval and consent from the participants. Thank you 

We have completed the STROBE guidelines checklist and uploaded to the manuscript centre. 

Funding sources have been added to the article body (also uploaded within the PLOS manuscript upload centre). See line 414. 

Our data dictionary will be uploaded to St George’s Fighsare. This has been noted in the relevant section of the manuscript submission platform and in the document (see line 426). 

Conflicts of interest and lack thereof are noted in the relevant section of the manuscript submission platform and in the document (see line 430).

---

## [Editor Report · Decision Letter 1]

27 Nov 2023

Seroepidemiology of SARS-CoV-2 in a cohort of pregnant women and their infants in Uganda and Malawi

PONE-D-23-25991R1

Dear Dr. Hookham,

We’re pleased to inform you that your manuscript has been judged scientifically suitable for publication and will be formally accepted for publication once it meets all outstanding technical requirements.

Kind regards,

Olatunji Matthew Kolawole, Ph.D.

Academic Editor

PLOS ONE

Additional Editor Comments (optional):

Required revisions have been well implemented
---

## [Editor Report · Acceptance letter]

16 Feb 2024

PONE-D-23-25991R1 

PLOS ONE

Dear Dr. Hookham, 

I'm pleased to inform you that your manuscript has been deemed suitable for publication in PLOS ONE. Congratulations! Your manuscript is now being handed over to our production team.

Kind regards, 

on behalf of

Dr. Olatunji Matthew Kolawole 

Academic Editor

PLOS ONE